# FLOW TREE: A DYNAMIC MODEL FOR NAVIGATION PATHS AND STRATEGIES

## ABSTRACT

Navigation is a dynamic process that involves learning how to represent the environment, along with positions in and trajectories through it. Spatial navigation skills vary significantly among individual humans. But what exactly differentiates a good navigator from a bad one, or an easy-to-navigate path from a hard one, is not well understood. Several studies have analysed exploration and navigation behaviour using static quantitative measures, like counts of positions visited or distance travelled. These static measures, however, are inherently limited in their ability to describe dynamic behaviors, providing a coarse quantification of the navigation process. To fill this gap, we introduce the *Flow Tree*, a novel data structure, which quantifies the dynamics of a group of trajectories through time. This is a discrete adaptation of the Reeb graph, a mathematical structure from topology, computed from multiple trajectories (from different people or the same person over time). Each divergence in trajectory is captured as a node, encoding the variability of the collection of trajectories. A Flow Tree encodes how difficult it will be to navigate a certain path for a group of humans. We apply the Flow Tree to a behavioural dataset of 100 humans exploring and then navigating a small, closed-form maze in virtual reality. In this paper we (1) describe what a Flow Tree is and how to calculate it, (2) show that Flow Trees can be used to predict path difficulty more effectively than static metrics, and (3) demonstrate that a trajectory through the Flow Tree is predictive of that individual's success. We (4) introduce a hypothesis testing framework over Flow Trees to quantitatively differentiate between the strategies of the best navigators from those of worst. Thus, we show that Flow Trees are a powerful tool to analyse dynamic trajectory data.[1]

## 1 INTRODUCTION

Navigation refers to the set of essential processes through which an individual adjusts one's position and orientation in space, familiarises one's self with new environments, and plans routes to specific destinations in familiar settings. Spatial navigation is a vital skill, as becoming disoriented or lost can lead to profound consequences.

At its core navigation is how agents move through spaces, both physical and abstract. It is a dynamic process that involves working out where one is, deciding where one wants to go, and figuring out how to get from the one place to the other. This dynamic process relies on cognitive and neural representations of the environment, positions within it, and potential routes. However, in humans, navigation abilities vary wildly. What is it about the underlying representations that leads to such different outcomes?

Despite its importance, we know little about the underlying representations supporting navigation and how they are acquired. Are they spatial grids, graphs, sequences of landmarks, or something else? Most current models rely on static representations (such as path averages or counts), yet real-world navigation involves a series of decisions made dynamically over time.

Flow Trees quantify the dynamics and variability of multiple trajectories over time, capturing divergences to assess the difficulty of navigating a path for both individual and groups of humans.

---

[1]The code will be made publicly available at [anon-github-link].

Figure 1: **Flow Tree: A dynamic model for navigation paths and strategies.** A Flow Tree encodes how difficult it is to navigate a path in a maze, by translating group cohesiveness into tree complexity. **A)** A group is given a navigational goal: get from A to B. **B)** Each navigator in the group then attempts to navigate that path, resulting in a trajectory composed of their sequence of decisions. All trajectories begin together at A, but only the successful ones end at B. **C)** The Flow Tree for this task is created using the times that the group splits in their navigation decisions. More splits mean more disagreements within the group. **D)** Then the Flow Tree can be used for exploratory analysis, to get a sense of path complexity. It can also be used to predict path difficulty and individual success, according to features, embeddings, or shapes.

In navigation research, the main limitation is not the quantity of data, but our ability to extract meaningful insights from it. Tools like DeepLabCut allow researchers to precisely identify animals' behavioral positions (Mathis et al., 2018), while virtual reality technologies have expanded our ability to gather rich navigation data for humans.

Most existing models rely on static representations, such as trajectory lengths (Richter & Duckham, 2008) or heat maps (Coutrot et al., 2022; Brunec et al., 2023), which provide only a coarse and imprecise description of the highly dynamic process. As datasets become more complex, composed of much more behavioural data over much long periods of time, like in Ward et al. (2024), new analytical tools are required to capture the complexity of the dynamic process. Flow Trees provide a complementary approach to testing scientific hypotheses in behavioural neuroscience, because they encode some of the spatio-temporal complexity of navigation.

In this paper, we (1) introduce the Flow Tree, a novel, dynamic way to analyse navigation. This data structure captures the complexity of a path by encoding the uniformity of group navigation decisions over time. We also show that this structure has predictive power, for both (2) the difficulty of a path for a group of individuals and (3) predicting an individual's success from their trajectory through the Flow Tree. Finally, we (4) propose and apply a hypothesis-testing scheme that uses Flow Trees to quantitatively compare different people, paths, and strategies. This allows us to move beyond static models and analyse the fluid, decision-making process of navigation more effectively.

## 2 RELATED WORK

### 2.1 MAZE NAVIGATION

This paper applies Flow Trees to behavioural analysis of navigation through discrete mazes. Brunec et al. (2023) collected a dataset of human navigation in a virtual-reality maze to explore how exploration behaviours affect cognitive map formation. They found that spending time in well-connected areas was predictive of success, but overall coverage was not. Gagnon et al. (2016) conducted a study where participants must navigate a virtual world to find a fixed set of objects as quickly as possible. They found that performance was linked to spatial memory ability. In a follow-up study Gagnon et al. (2018) found gender differences in free exploration in terms of navigational efficiency and location visit counts. Ward et al. (2024) looked at another virtual navigation task, split into exploration and then test phases. They investigated how 11 different exploration metrics, along with survey data, might affect how accurate and efficient individuals are in the test phase. Dynamic frameworks like the Flow Tree provide an additional strategy to encode and analyse behaviour over time.

## 2.2 NEUROSCIENCE AND NAVIGATION

Research into the neural basis of navigation seeks to understand how the brain creates and then uses spatial representations, like cognitive maps (Whittington et al., 2022), in order to guide behaviour. Navigation behaviour is supported by a distributed neural network, which integrates spatial, sensory and motor information (Taube, 2007). The dynamic nature of navigation—learning new spaces, requiring a continuous feedback loop to factor in environmental changes, and accounting for both past and future states—makes neural representation incredibly complex. In order to fully understand how neural systems produce behaviour, we must find a way to link dynamic models of neural systems with models of behaviour.

## 2.3 REEB GRAPHS

Navigation behaviours cannot be adequately explained with Euclidean distances and angles alone (Warren, 2019). Cognitive maps may be best modelled as labelled graphs, with local and approximate metric information (Chrastil & Warren, 2014; Warren et al., 2017).

The Flow Tree is a directional variant of another type of graph, a Reeb graph. This is a data structure in topology that represents how the level sets of a scalar function over a manifold evolve (Reeb, 1946). The nodes of a Reeb graph correspond to places where components of the level set split or merge, and its edges correspond to the continuous components between those points. These have been applied to shape analysis (Biasotti et al., 2008) and data skeletonisation (Ge et al., 2011). Recently, Reeb-graph-based methods have also been applied to analyse spatiotemporal data, such as motion tracking and trajectory analysis. For example, Zhang et al. (2024) proposed a Reeb-graph based model to identify anomalies in a dataset of human GPS trajectories. Similarly, Flow Trees track how groups of trajectories evolve over time, but they only encode when groups split, ignoring when they merge again. That there are no cycles helps to encode time in the tree, where moving down the tree from root to leaves means moving forward in time.

## 3 METHODS

We introduce the Flow Tree to study navigation data (see Figure 2). This is computed from collection of a group of people's attempts to navigate from a starting location to a common goal. Once built, we use Flow Trees to extract new insights from navigation data, about the difficulty of a navigation task for the group, as well as an individual's relative performance. To this aim, we introduce exploratory and inferential statistics, along with machine learning approaches, for Flow Trees. We consider three complementary perspectives: extracting features from the trees, embedding the trees in a latent space, and studying the shapes of the trees.

### 3.1 BUILDING FLOW TREES

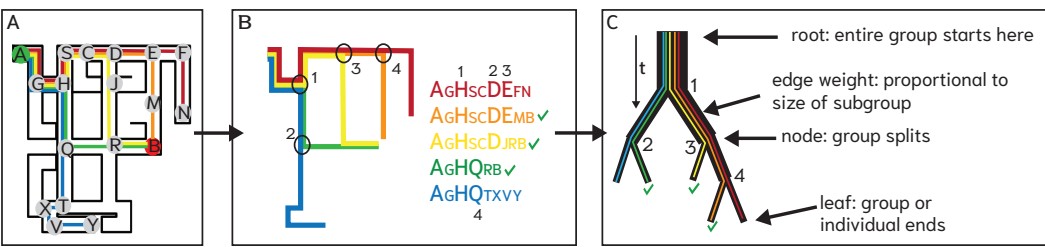

Figure 2: **How to build a Flow Tree.** Flow Trees are computed by collecting a group of trajectories, which start at one location and attempt to get to another. **A)** Collect group's trajectories from A to B. **B)** Identify group splits. **C)** Construct the Flow Tree, starting at the root node, with each trajectory terminating in a leaf. A Flow Tree node is created whenever the group does not make a unanimous next decision (i.e., when the group splits in space and time). Edges and their weights correspond to the sizes of the subgroups created by this split.

We compute a Flow Tree for a certain navigation goal or path (i.e., to get from point A to B) by looking at a collection of attempts, or trajectories. This Flow Tree represents how well a group sticks together in space through time, as they navigate, because it contains information about a group's cohesiveness and therefore the path's difficulty. An easy-to-navigate path creates a simpler Flow Tree, because any given group makes more similar decisions, as individuals make fewer errors. A Flow Tree is an attributed tree.

**Definition 1** (Graphs and Trees). A *graph* is an ordered pair $G = (V, E)$ where $V$, a set of nodes and $E \subseteq \{\{x, y\} \mid x, y \in V \text{ and } x \neq y\}$, a set of edges, which are unordered pairs of nodes. A *tree* $T = (V_T, E_T, r)$ is a graph in which any two nodes are connected by exactly one path, and where one node $r$ is designated as the root. The edges of the tree are then assigned a natural orientation, going away from the root $r$. An *attributed tree* is a tree $T$ with an attribute map $a : E_T \mapsto \mathbb{R}$ which assigns an attribute $a(e)$ to each edge $e \in E_T$, for example, an edge weight. A tree can be equivalently represented by its *adjacency matrix*, defined as the square $V_T \times V_T$ matrix $A$ such that the element $A_{ij}$ is $a(e)$ when there is an edge from vertex $i$ to vertex $j$, and zero otherwise.

To compute this (pseudocode in Appendix C, Algorithm 1), we select a start and end node (e.g., A and B in Figure 2). Then we collect a group of people's trajectories as they attempt to navigate from start to end. The start node must be the same for every trajectory in the collection, whereas the trajectories' end nodes can vary. Because all trajectories start in the same place, the group starts as one at the root node of the Flow Tree. From here, a Flow Tree node is added at points when the group splits (i.e., when people in the group make different decisions). In this example, the second tree node would happen at the third step in the trajectory, when the first three people (red, orange, yellow) go from H to S and the other two (green, blue) go from H to Q. The Flow Tree is labelled with the last point at which the group's decision was together (i.e., H). A leaf node is added when a trajectory or group of trajectories ends. The edges of the tree are weighted by the fraction of the original group of people who belong to the group making that decision / passing along that trajectory. This means the edge from A to H has a weight of 1, the edge from H to D a weight of $\frac{3}{5}$ and from H to Q a weight of $\frac{2}{5}$.

**Definition 2** (Flow Tree). Let be given a graph $G = (V, E)$ representing a navigation map where intersections are nodes in $V$ and corridors are edges in $E$. Take $A, B \in V$ two nodes in the map and consider $M$ trajectories going from $A$ to $B$. The *Flow Tree* is defined as the attributed tree $T = (V_T, E_T, r, a)$ with root $r = A$, node set $V_T \subset V$ and edge set $E_T$ built according to the procedure described above, and attribute map $a : E_T \mapsto \mathbb{R}$ assigning the number of trajectories going through each edge $e_T \in E_T$.

Because Flow Trees are only concerned with group splittings, they are less sensitive to path length and net group size (i.e., related more to the topology). (Another collection of paths that would generate an equivalent Flow Tree to {ABDG, ABCI, ABCI} would be {AXYZBDG, AXYZBCI, AXYZBCI}.) If the path is very easy and the group's navigation decisions are uniform the Flow Tree will be very simple. If the path is complicated, or the group makes random decisions, the Flow Tree will be complex. Longer paths do offer more of a chance for a group to fracture, and a smaller group will give a coarser Flow Tree and therefore a less statistically robust estimate of the path complexity, but the high-level pattern should be the same.

Navigating a path involves making a series of decisions over time, moving in a direction in both space and time. The tree structure of Flow Trees encodes the dynamics inherent in this. Each depth level in the Flow Tree is a new level in time. Decisions made earlier in the navigation are encoded closer to the root Flow Tree node. Navigation can involve loops in space (particularly if one is lost), but Flow Trees permit no cycles. Creating a Flow Tree, therefore, effectively unrolls the trajectories to create branches flowing from root to leaves. Static spatial representations, like heat maps, might show uniformity of group navigation in space, but it collapses over the time dimension, effectively ignoring the sequential nature of the navigation process. Critically, Flow Trees allow us to consider the temporal dynamics of group travel.

### 3.1.1 COMPLEXITY

Computing a Flow Tree involves iterating over a group trajectories, each of which goes through a sequence of nodes.

**Proposition 1** (Time and Space Complexity). *Consider a Flow Tree built from $M$ trajectories, the longest of which traverses $N$ nodes.*

- *The worst-case time complexity of the tree generation is $O(N * M)$.*

- *The worst-case space complexity of tree is $O(M)$.*

For our recursive implementation, the worst-case time complexity is $O(N * M)$. To compute the Flow Tree we loop over each node in the trajectory, so up to depth $N$. At each step, we loop through each trajectory to check whether all decisions taken at that node are identical or not. Using hashing this takes $O(M)$. Creating a node and edge in the Flow tree is $O(1)$. When the group splits, we still iterate through each node (just in separate groups); it might be one has $M - x$ and the other $x$ trajectories, which will give $O(M - x + x) = O(M)$ for each of the $N$ steps.

The worst-case for space complexity is when the group bifurcates at each of the $N$ steps in the $M$ trajectories. The tree will start with 1 root node and end up with $M$ different leaf nodes. To make this simpler, let's assume M is a power of 2 and $N = \log_2 M$ (longer trajectories will not matter for space, as it is differing decisions among people that create Flow Tree nodes). The total number of nodes then is $\sum_{0 \leq n \leq N} M/2^n = 1 + 2 + 4 + ... + \frac{M}{4} + \frac{M}{2} + M = 2M - 1 = O(M)$.

## 3.2 USING FLOW TREES TO ANSWER SCIENTIFIC QUESTIONS

Flow Trees are powerful tools for analysing navigation data. They can be used to better understand and quantify why some paths are more difficult than others, dynamically predict an individual's success, and quantify different group and exploration dynamics. In this subsection we describe three different ways to extract information from them: features, embeddings, and shapes.

### 3.2.1 ANALYSE TREE FEATURES

As a first pass, we can use standard tree features to quantify them for purposes of statistics and machine learning. The features that we consider are:

- **Number of Nodes and Edges:** Nodes and edges are highly correlated. A higher number of nodes indicates more frequent divergence in the group's trajectories, as each node represents a decision point where trajectories split.

- **Tree Depth:** The maximum number of decision points along any trajectory, corresponding to the longest path from start to end. This is the maximum amount of times along a trajectory that different decisions were made, bounded above by the longest trajectory. A smaller depth compared to the ideal path length suggests consensus on certain correct decisions, indicating easier sections of the path.

- **Average Node Degree:** The average number of branches (edges) per node, measuring the level of divergence at each decision point. A higher average node degree indicates more possible choices at each split, signaling greater variation in the group's trajectory decisions. This depends on the maze's structure (or the degree of the nodes in the maze).

- **Diameter:** The longest distance between two leaf nodes, representing the maximum divergence between any two individuals' trajectories. A larger diameter suggests a greater spread of possible paths and perhaps more challenging navigation.

- **Number of Leaves** The total number of leaf nodes. Fewer leaves mean there are more people who take the same exact trajectory as others.

- **Average Path Length** The average distance from the root node to a leaf node, indicating the typical number of decisions made before reaching an endpoint.

Then, we apply standard techniques for exploratory data analysis, inferential statistics, and machine learning on these tree features. It is important to note that some features are discrete (e.g., number of leaves), others continuous (e.g., average node degree), and that they have different units. Therefore, in our analyses, we standardize the features and use methods capable of handling this diversity in data types. In practice, we use Python libraries such as scikit-learn (Pedregosa et al., 2011).

### 3.2.2 ANALYSE TREE EMBEDDINGS

Another way to use trees quantitatively is to embed them in a high-dimensional latent space, such that similar trees are close together in that space. Existing approaches of graph embeddings can be applied to trees which are graphs without cycles. While there exists a diversity of graph-embedding methods, our analysis will focus on the graph2vec algorithm (Narayanan et al., 2017). This embedding technique is trained in an unsupervised manner, is task agnostic, and resulting vectors in latent space capture structural equivalences between graphs.

Next we apply standard techniques for exploratory data analysis, inferential statistics, and machine learning on these tree embeddings. It is important to note that the results of these analyses also depend on the quality of the embedding model and the number of dimensions of the latent space. A hyperparameter search is necessary. Additionally, results might be harder to interpret than results from the approach with the tree features. In practice, we use the Karate-Club Python package (Rozemberczki et al., 2020) for the graph embeddings and scikit-learn for the subsequent statistics and machine learning analyses (Pedregosa et al., 2011).

### 3.2.3 ANALYSE TREE SHAPES

Tree features only capture a fraction of the information present in the Flow Tree. Embeddings might also collapse information, and additionally depend on the choice of embedding algorithm and dimension of latent space. Consequently, we introduce a third alternative that analyses the shapes of the trees. This approach does not collapse any information about the Flow Tree except the indexing of its nodes, which is information that does not hold scientific meaning. As a drawback, it comes at an added conceptual and computational complexity compared to the two previous methods.

We use the formalism introduced by Calissano et al. (2020) for shapes of graphs, but for the Flow Trees. The shape of the Flow Tree is defined as the adjacency matrix of the Flow Tree after we remove any information related to the indexing of the nodes. Mathematically, the shape $[T]$ of the Flow Tree $T = (V_T, E_T, r)$ with adjacency matrix $A$ is defined as:

$$[T] = \{PAP^T \mid P \in \mathcal{P}_N\}, \tag{1}$$

where $N = |V_T|$ the number of nodes in the tree, and $\mathcal{P}_N$ the set of permutation matrices. In other words, the shape of the tree $T$ is the set $[T]$ of adjacency matrices that we obtain by re-indexing the nodes of $T$. If we have a dataset of Flow Trees with different number of nodes, we pad the smaller adjacency matrices with 0's. This will allow us to compare matrices of similar sizes for the subsequently statistics and machine learning analyses.

Then, we apply techniques of exploratory data analysis, inferential statistics, and machine learning on these tree shapes. However, the tree shapes are complicated mathematical objects in that they are not vectors – in contrast to tree features and tree embeddings. Mathematically, the space of tree shapes is not a vector space, but a so-called *metric space*, i.e., a set that has a notion of the distance $d([T], [T'])$ between every pair of elements $[T], [T']$. This distance is defined as followed:

$$d([T], [T']) = \inf_{P \in \mathcal{P}_N} \|A - PA'P^T\|_2, \tag{2}$$

for $A, A'$ the adjacency matrices of $T, T'$ respectively, and $\|.\|_2$ the Frobenius distance between matrices. Intuitively, the distance between two tree shapes is defined as the minimum distance between their adjacency matrices, while going over all possible combinations of node indexing for the second tree. Equipped with a notion of distance, we apply statistics and machine learning approaches to the space of tree shapes Calissano et al. (2020). In practice, we use the Python package geomstats (Miolane et al., 2020) to compute the tree shapes and their distances. Then, we employ machine learning and statistical analyses methods that solely rely on the notion of distances between data points.

## 4 EXPERIMENTS

### 4.1 MAZE DATASET

The Maze dataset Ward et al. (2024) comes from a study on navigation with data from 100 healthy young adults (47 female, ages 18-37, $M = 20.7$, $SD = 3.13$). There is a maze of tall hedges

constructed in virtual reality, with nine target objects and four paintings, which serve as landmarks. At any given time, a participant can move forward, turn right, or turn left. Participants first spend a total of 16 minutes exploring and orienting themselves in the maze. And then they are given 48 time-limited trials ($\frac{2}{3}$ of the $^9P_2$=72 possible test paths), where they are dropped at one object and asked to navigate to another. During this test phase, objects are replaced by red balls to prevent additional learning. The behavioural data we use consists of a temporally extended list of each participant's navigation decisions during the test phase.

## 4.2 VISUALISATION: EXPLORATORY STATISTICS ON FLOW TREES

Flow Trees are a powerful tool for exploratory analysis. They provide an intuitive way to look at aggregated human navigation data to get a sense for which paths might be harder, which might be easier, and why (see Figure 3).

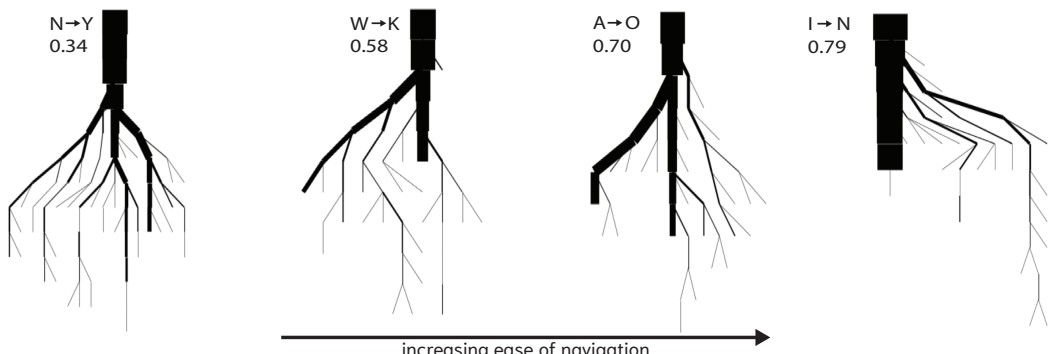

Figure 3: **Visualising path complexity with Flow Trees.** Selected test-path Flow Trees, for start-end combinations of N/Y, W/K, A/O and I/N; subject success rate for each path shown. Flow Trees for more difficult paths (left) are more complicated and spindly, with lot of branching. Flow Trees for easier paths (right) have a more clearly defined backbone, indicating that many people made the same (correct) decisions. In the cases where two clear backbones emerge, there are two equivalent minimal paths. See Figure A.6 in Appendix A for all 72 start-end combinations.

## 4.3 MACHINE LEARNING ON FLOW TREES

We show that Flow Trees are a powerful representation of navigation data for machine learning tasks. To demonstrate this, we consider the regression task of using Flow Trees to predict path difficulty in three ways: using tree features, embeddings, and shapes. We compare them to two baselines: a set of semantically significant static features (Euclidean distance of the path, the distance along the minimal path, the average number of nodes, distance, and turns traversed by participants) and a heat map (counting how many times each node has been traversed by people in the group). We use linear regression when inputs belong to a vector space. Because shape is defined only using a distance metric, we use k-nearest-neighbors regression to compare across all representations. Using Linear Regression and KNN Regression (see Figure 4) we find that Flow Tree features are the best predictor of path difficulty.

One difficulty for static metrics and heat maps is directionality. For instance, many more people successfully navigate from N to Y than they do from Y to N. A difference here will not be captured in path length, or number of nodes traversed. Instead, the difference lies in how those nodes are dynamically traversed. This may contribute to Flow Trees' superior performance.

## 4.4 FLOW TREES AS DYNAMIC PREDICTORS

Flow Trees contain information about how difficult a navigation task is for a group of people. They also allow us to predict how well an individual will perform at a given task, based on their trajectory through the Flow Tree. Crucially, we can use information encoded in the structure of a Flow Tree

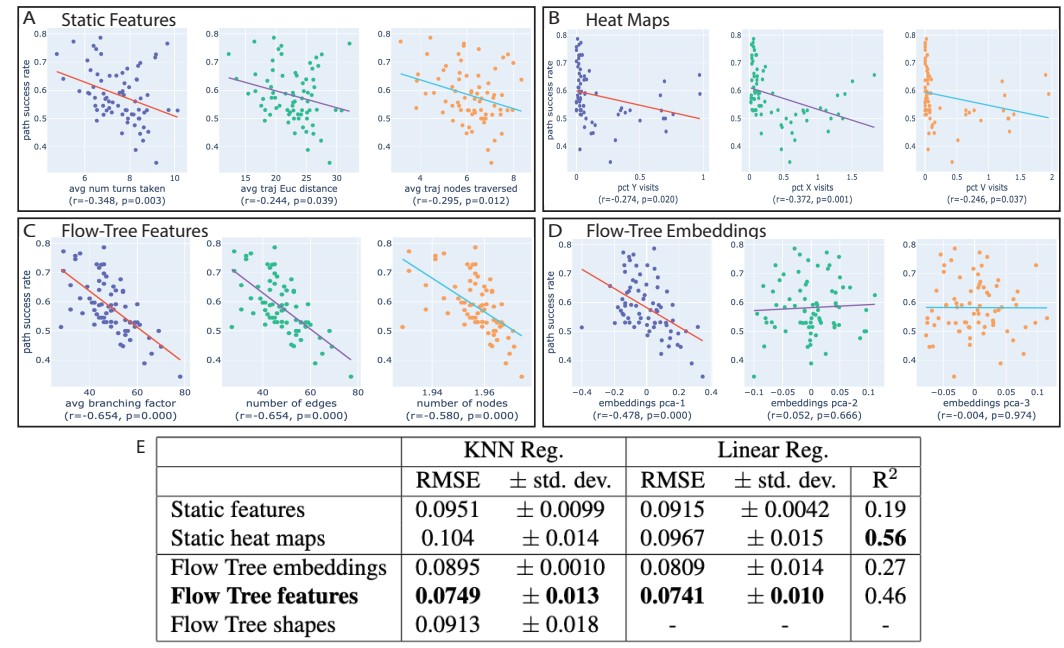

Figure 4: **Flow Trees encode path difficulty.** We compare static features and heat maps and Flow Tree features, embeddings and shapes as predictors of path difficulty (the percentage of participants that successfully navigate a test path), using 5-fold cross validation. We first test correlations of all linear-regression features and path difficulty across all 72 Flow Trees, one for each start-end combination. Shown are the most significant three features for each feature set, along with their Pearson correlation coefficient and associated p-value. **A)** Static features: average number of turns taken by each participant, average Euclidean distance travelled by each person, and average number of maze nodes traversed. All are significantly inversely correlated with path success rate, or positively correlated with path difficulty. **B)** Static heat maps (from the heat map, we extract the number of times each node is visited): proportion of visits to Y, X, and V. All three are correlated with path difficulty. **C)** Flow Tree features: average branching factor, number of edges and nodes. These are all significantly and strongly negatively correlated with path success rate. **D)** Flow Tree embeddings: the top three principle components for 16-dimensional graph2vec embeddings. Only the first one is significant, suggesting that most of the information used in Flow Trees might be collapsible to fewer dimensions. **E)** We find that Flow Trees are predictively powerful, with significantly lower RMSE than the static baselines.

to form dynamic predictions about an individual's likelihood of success or failure before they finish navigating. The Flow Tree is a structure that can be trained on human behaviour in order to predict the behaviour of future humans doing the same task.

We do this as follows (see Figure 5): First, we separate the trajectories for each test path into two groups (70/30). We use the first group's trajectories to construct a Flow Tree. Then, for each of the trajectories in the second group, we identify which Flow Tree nodes that trajectory would pass through on the way from root to leaf. At each of the nodes, we can compute a maximum likelihood estimate by counting the number of successful and unsuccessful trajectories belonging to the subtree of that node. This number is the likelihood at that level in the tree that the individual will successfully complete the path.

As an individual trajectory flows down from root to leaf, the accuracy of prediction increases (see Figure 5). At the first level, after one split, the average AUC across all test paths is 0.6. And then, at the three level, after only three splits, the average AUC is 0.81, indicating good predictive power.

The success of this sort of dynamic Flow Tree prediction depends on the path being navigated. For some start-end combinations, after one split, Flow Trees have almost perfect prediction of success;

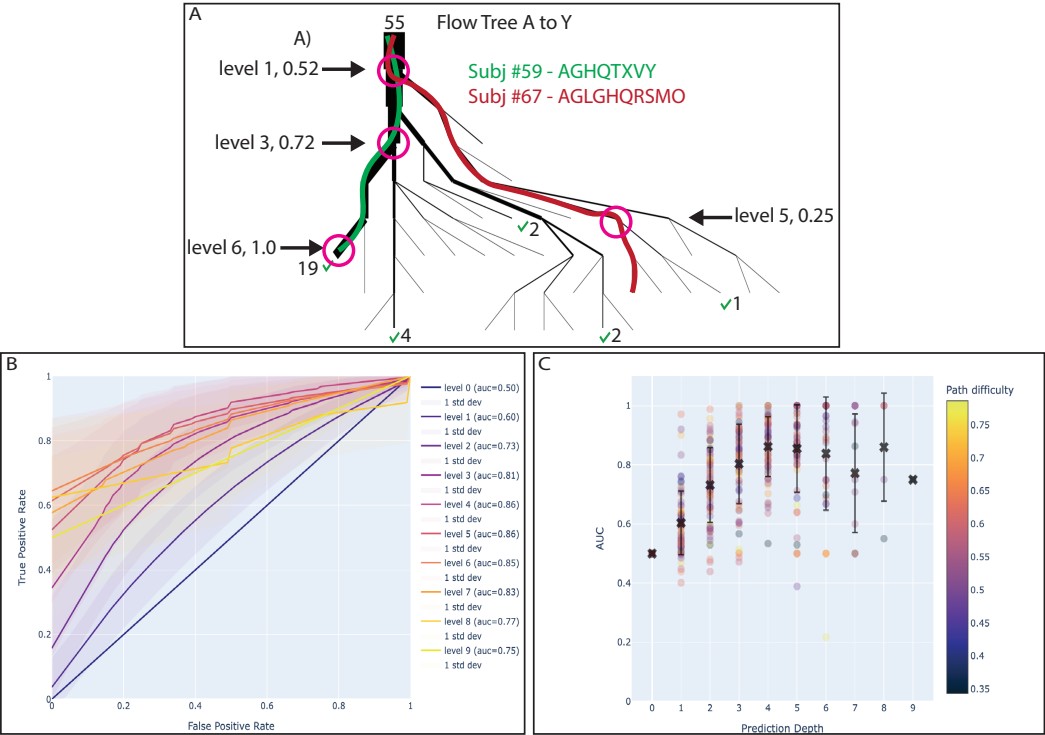

Figure 5: **Dynamic prediction using Flow Trees.** The success of an individual trajectory for a given goal can be predicted by following that trajectory through a Flow Tree. **A)** A Flow Tree is constructed from the trajectories of 55 individuals trying to get from A to Y. We use the Flow Tree to follow the trajectories of two participants not in that group, #59 (who got it right) and #67 (who got it wrong). At each level in the Flow Tree, we compute a likelihood that an individual trajectory at that node will get it right. **B)** Mean ROC curves over all combinations of start/goal to demonstrate the utility of these predictions, using 10 different train-test splits for each start/goal combination. Progressing down the tree yields more accurate predictions. **C)** AUC of the ROC curve for each level for each different start/goal combination.

for others, they lack consistent ability to predict through all levels. This does not depend on the overall success rate of participants on the path.

## 4.5 INFERENTIAL STATISTICS ON FLOW TREES

We define hypothesis tests to determine whether two groups differ in their Flow Trees. Then we apply this to answer the following question: do Flow Trees computed from the trajectories of successful navigators differ from those of less successful ones? We chose this group split on purpose as a proof-of-concept in order to validate this method. The answer should be yes, because groups of more successful navigators should make more uniform and correct decisions, leading to simpler Flow Trees.

**Hypothesis-test definition**   Flow Trees are represented either as features, embeddings or shapes. The first two are vectors of scalars, but shapes do not have a quantitative value to represent them directly. They do, however, have well-defined distances between them. Therefore, we consider Flow Trees as elements of a metric space, where the distance metric is the shape distance (for tree shapes) and the Euclidean distance (for tree features and tree embeddings).

Suppose that the observed Flow Trees consist of independent samples $\left(T_1^X, \ldots, T_m^X\right)$ and $\left(T_1^Y, \ldots, T_n^Y\right)$ divided into two groups, for example $m$ Flow Trees formed from successful nav-

igators and $n$ from unsuccessful ones. Assume the number of individuals in each group are the same to not inherently bias the complexity of the tree (larger groups of trajectories give the opportunity for more split decisions). The objective is to test whether the Flow Trees are drawn from the same distribution. The null hypothesis $H_0$ states that the samples are generated from the same probability distribution.

$$H_0 : \left(T_1^X, \ldots, T_m^X\right) \text{ and } \left(T_1^Y, \ldots, T_n^Y\right) \text{ are drawn from the same distribution.} \tag{3}$$

To perform the test, we define the test statistic $D$ as the distance between the means of the groups. On the observed trees, the test statistic for shapes takes the value $D^{\text{obs}}$ given by:

$$D^{\text{obs}} = d(\hat{\mu}^X, \hat{\mu}^Y), \tag{4}$$

where $d$ is the distance metric of the metric space and $\hat{\mu}^X, \hat{\mu}^Y$ are the Fréchet means of group $X$ and $Y$ respectively. In a metric space, the Fréchet mean is defined as the point that minimizes the sum of the squared distances to the data points (Fréchet, 1948). Thus, $D$ only relies on the notion of distance and naturally applies to our general framework of metric spaces that encompasses the representation of Flow Trees as tree features, embeddings or shapes.

We can then apply a permutation test to the data (defined more formally in Appendix B. We reject the null hypothesis $H_0$ at significance level $\alpha$ if $p < \alpha$. In that case, we conclude that the two samples of Flow Trees are generated from statistically different distributions. Otherwise, we fail to reject the null hypothesis. This would mean that there is no significant difference between the means, and the groups' Flow Trees are the same.

**Hypothesis-test application**   We first split the participants into two groups, based on their average accuracy during test trials. They we compute two Flow Trees for each start-end combination, yielding $72 * 2 = 144$ Flow Trees in total. We can then compute the features, embeddings and shapes of these Flow Trees. The null hypothesis in this case is that all the Flow Trees are the same. We run a permutation test looking separately at Flow Tree features (0.448, $p = 0.002$), embeddings (-4.875, $p = 0.002$) and shapes (8.08, $p < 0.001$). We reject the null hypothesis in all cases, indicating as expected that there is a difference in how these two groups navigate.

## 5   CONCLUSION

Flow Trees dynamically predict individual success; their features, embeddings and shapes predict of path difficulty; and they can be used to test hypotheses about groups of people or trajectories.

Hypothesis testing is particularly interesting, because it allows one to ask questions about group navigational dynamics. Males are more successful at navigating than females (Munion et al., 2019). Are the strategies of males different from those of females? What about divisions according to video-game experience, age, or handedness? By controlling for overall accuracy, Flow Trees can help to uncover differences in the dynamics of group navigation, beyond just success rates.

Flow Trees can also represent an individual's exploration strategy by collecting a set of trajectories by chunking the decisions they make during a prolonged exploration phase. This lets one test hypotheses like: exploration strategy changes after one has seen all of the objects, or the strategies of successful navigators differ substantially from those of less successful ones.

Broadly applicable across spatial navigation datasets–whether in discrete mazes or continuous spaces–Flow Trees can also be applied to abstract graph spaces, like the internet or Wikipedia. Their branching structure captures the complexity of paths, highlighting efficiency, node difficulty, and the shared or divergent experiences of users. The Flow Tree's geometry provides valuable insight into the structure of any navigable environment.

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

# A APPENDIX FIGURES

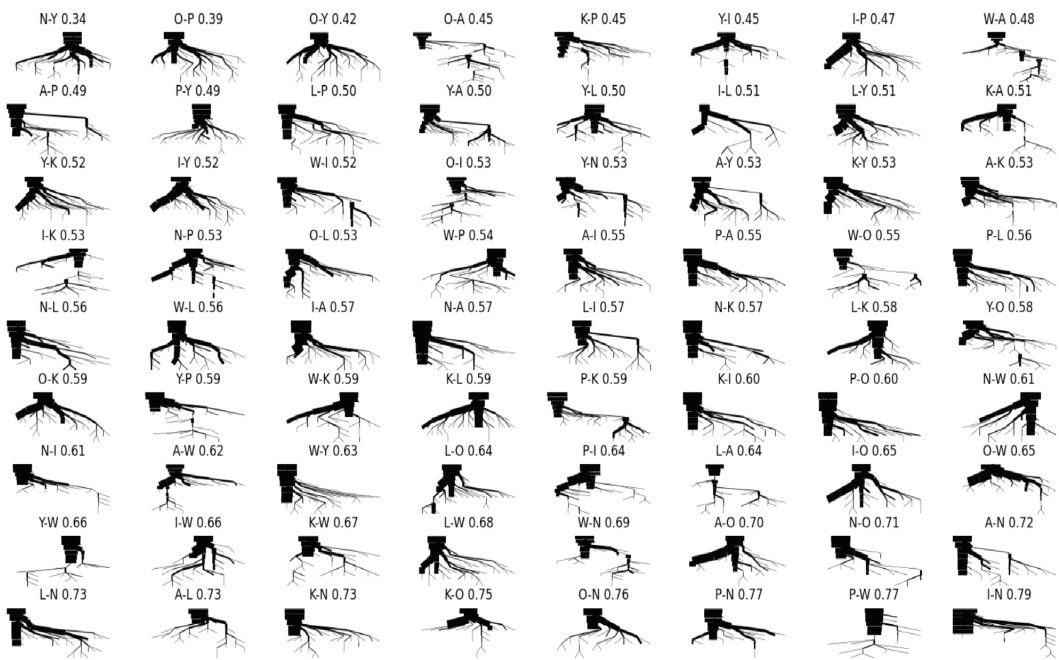

Figure A.6: **Visualising path complexity with Flow Trees.** Shown are the Flow Trees for all 72 test paths in the maze dataset. They are sorted by difficulty of success (according to the percentage of people who correctly navigated from start to end). The Flow Trees for the more difficult paths (towards the top) are more complicated and spindly. They have a lot of branching. The Flow Trees for the easier paths (towards the bottom and right) have a more clearly defined backbone, indicating that many people made the same (correct) decisions. In the cases where two clear backbones emerge, there are often two equivalent minimal paths.

# B APPENDIX DEFINITIONS

Under the null hypothesis $H_0$, the observed trees can be permuted or randomly assigned to either of the two groups without altering the distribution of the data. More formally, combine the data as $T^Z = \left(T^Z_1, \ldots, T^Z_N\right) = \left(T^X_1, \ldots, T^X_m, T^Y_1, \ldots, T^Y_n\right)$ where $N = m + n$. Let $\pi = (\pi(1), \ldots, \pi(N))$ represent a permutation of $\{1, \ldots, N\}$. The permuted data $T^Z_\pi$ are given by $T^Z_\pi = \left(T^Z_{\pi(1)}, \ldots, T^Z_{\pi(N)}\right)$. Under the null hypothesis $H_0$, the random variables $T^Z$ and $T^Z_\pi$ follow the same distribution.

Then, for each permutation $\pi$ of the data, we recompute the means $\hat{\mu}_\pi^X, \hat{\mu}_\pi^Y$ and their distance $D_\pi$. This provides a distribution of distances $\{D_\pi\}_\pi$ that we compare to the observed value $D^{\text{obs}}$. The p-value of the test is calculated as the proportion of permutations $\pi$ where the distance between the means is greater than $D^{\text{obs}}$:

$$p = \Pr(D_\pi > D^{obs} \mid H_0 \text{ is true}). \tag{5}$$

## C   APPENDIX CODE

---
**Algorithm 1** Flow Tree Generation

---
1: **procedure** GENERATEFLOWTREE(start_node, trajectories)
2:     *# Base Case*
3:     **if** all_traj_are_same **then**
4:         add_leaf_node, add_edge
5:     *# Recursive step*
6:     **for** next_nodes in trajectories **do**
7:         **if** all_next_nodes_same **then** Continue
8:         add_node(last_node_together)
9:         traj_subset $\leftarrow$ split_into_subgroups(trajectories)
10:         **for** traj_subset in traj_subsets **do**
11:             add_edge(prev_node, new_node, group_size)
12:             GenerateFlowTree(new_node, traj_subset)

---

