# OpenReview forum: "Flow Tree: A dynamic model for navigation paths and strategies"
_ICLR.cc/2025/Conference — ICLR 2025 Conference Withdrawn Submission_

### Official Review · Reviewer_bFEP · 2024-10-21

**Soundness:** 3
**Presentation:** 3
**Contribution:** 1
**Rating:** 3
**Confidence:** 3

**Summary:**

This work presents Flow Tree, a data structure which quantifies the dynamics of a group of trajectories over time. The paper demonstrates how these trees can be useful for statistical hypothesis testing on groups and for determining success likelihood of a partial trajectory.

**Strengths:**

- Overall the paper is well written with detailed statistical analysis.
- The paper includes helpful visualizations, such as Flow Tree diagrams

**Weaknesses:**

The main shortcoming of this paper is that the data structure being proposed is not a particularly novel one. It is similar to a prefix tree (or trie) data structure.

After a quick search, I was able to find a few works that use prefix trees (trie) for synthesizing trajectory data in applications such as traffic, GPS data, and higher-level decision making:

- Kuo, C.-T., et al. (2018). *On The Equivalence of Tries and Dendrograms - Efficient Hierarchical Clustering of Traffic Data*.
- Wang, N., et al. (2024). *DPTraj-PM: Differentially Private Trajectory Synthesis Using Prefix Tree and Markov Process*.
- Sitanskiy, S., et al. (2023). *Learning and Recognizing Human Behaviour with Relational Decision Trees*.

Many others discretize continuous trajectories into graph structures instead of trees. I found the following to be a good summary of the existing data structures and learning methods for trajectory datasets: Graser, A., et al. (2023). *Deep Learning From Trajectory Data: A Review of Deep Neural Networks and the Trajectory Data Representations to Train Them*.

One such example is the Reeb graph paper cited in the related work section. Although the authors discuss the differences between the Reeb graph and Flow Trees, they never compare to the reeb graph (or any other graph data structure) in their statistical analysis.

In the section titled In the section titled “flow trees as dynamic predictors”, the authors show that flow trees can be a good indicator of the likelihood of task success, but they didn’t show that they are a better indicator than other representations.

No code or supplementary material was provided, which limits reproducibility.

Some minor points:

Figure 4 is a bit disorganized with overlapping text, different font sizes and shapes, and mismatched color schemes. Also, the table should be made a latex table rather than a picture of a table.

(Page 10) ”They we compute two Flow Trees” — They → then

**Questions:**

Why only consider Y, X, and V in the heatmap regression?

Wouldn’t deviation from an optimal path also be a good predictor of task success from a partial trajectory?

“One difficulty for static metrics and heat maps is directionality. For instance, many more people
successfully navigate from N to Y than they do from Y to N.” Why not use a bidirectional heatmap then?

In the section titled “Machine learning on flow trees”, why try and predict path success rate from dataset-level statistics? Isn’t is easy to compute path success rate from the trajectories? What is a situation where you’d have a flow tree but couldn’t compute the success rate?

---

### Official Review · Reviewer_VmTV · 2024-11-07

**Soundness:** 3
**Presentation:** 3
**Contribution:** 2
**Rating:** 6
**Confidence:** 3

**Summary:**

This paper introduces the Flow Tree, a novel data structure that quantifies the dynamics of a group of trajectories through time. Suppose we have a dataset of a group of trajectories navigating some maze, provided by a group of human participants. Then, this paper proposes to represent it as a flow tree. Each split of the flow tree corresponds to a split of the trajectory group. So intuitively, if all trajectories are similar, the flow tree will have very few splits. Essentially, the flow tree ignores the spatial information and lengths of the group trajectories but focuses on representing the group splits. This paper also develops methods to construct flow trees and use flow trees for various potential applications.

To analyze flow trees, the paper proposes 3 tools:
1. Tree features, such as number of nodes and edges, tree depth, average node degree, diameter, number of leaves, and average path length.
2. Embedding: one can embed the flow tree to a low-dimensional vector.
3. Tree shape: the authors build on Calissano et al. (2020) to define a metric space of flow trees, with a metric measuring the difference between two flow tree shapes.

To use flow trees, the paper conducts one experiment using flow trees to analyze the dataset of trajectories of humans navigating a maze.
1. Use a flow tree to predict path easiness (how many people arrive at the goal location in the maze). The authors compare tree features, embedding, and shape against two baseline methods. Tree features perform the best.
2. Use a flow tree to predict how well an individual will perform at a given task based on their trajectory in the past. The authors demonstrate this application.
3. Use a flow tree to test whether two groups of trajectories come from the same distribution or not. The authors demonstrate this application.

**Strengths:**

- The problem is well-motivated.
- This paper is well written.
- The proposed framework can have many potential applications.

**Weaknesses:**

- For the hypothesis testing example, it seems that the authors split the dataset into 2 groups based on average accuracy, and the flow-tree-based test rejects the hypothesis that the 2 groups are drawn from the same distribution. This nicely illustrates the message, but it is a bit contrived.
- The paper's results rely on data of 2D paths in a maze, where it is clear when a path splits into two paths. However, in the real world, for example, pedestrian paths may never coincide, so it is hard to define what a path split is. It would be great if the authors could discuss the potential challenges in using a flow tree for analyzing real-world human navigation data.
- Flow tree depends on the map. In other words, in a single-road map, all paths are the same, resulting in a very "dense" flow tree. On the other hand, on a very complicated map, suppose that the humans are all experts. As a result, all their paths are very similar to each other, resulting in a very "dense" flow tree. It seems that flow trees have a hard time distinguishing between these two cases. If this makes sense, it might be helpful to mention this dependence of flow trees on the map.

**Questions:**

- Flow tree seems to only work when all the trajectories start at the same point. What if the trajectories begin at different locations? I am curious about whether it would fundamentally change the problem.
- In Fig.4, flow tree shapes perform much worse than flow tree features. Flow tree shapes perform similarly to the baseline method - static features. I am curious about why this is the case. Perhaps the flow tree shape characterizes the adjacency matrix, which is not so relevant to the easiness.
- In Fig.4, flow tree embeddings are not as well as flow tree features. I wonder whether the embeddings are interpretable. Perhaps it could be interesting to visualize the embeddings to understand more about why it does not perform as well as flow tree features.
- This paper considers trajectory datasets for mazes. However, if the trajectories contain noise, how would the flow tree method handle noise?

---

### Official Review · Reviewer_ZcVa · 2024-11-08

**Soundness:** 2
**Presentation:** 3
**Contribution:** 1
**Rating:** 3
**Confidence:** 3

**Summary:**

The authors propose "Flow Tree", a topological representation of human navigation. The authors propose to use this to evaluate complexity of navigation problems, predicting success in navigation etc. The method is evaluated on data from a maze experiment with 100 people.

**Strengths:**

- Human navigation behavior is not well understood so any insights there could be valuable
- The paper is well-written and mostly easy to read.

**Weaknesses:**

- The "Flow Graph" representation appears to me to be an extremely simplified view of human navigation behavior, basically just capturing *when* during navigation that people pick a different path at an intersection.
- The experiment is also extremely simple, appearing to be a Maze with clear discrete paths. How would this generalize to e.g., a forest with a mix of open areas and heavy vegetation?
- The problem setting seems to ignore the perception part of navigation problems. In the robotics and computer vision literature there is extensive literature on mathematical methods for navigation (e.g., SLAM [1*]), and what makes these problems hard (e.g., similar areas). This is ignored.
- I am not sure if this approach is really of interest to the ML community. It might be of interest in cognitive science as it focuses on modeling human behavior. However, subject to the limitations above, I am not entirely convinced that this model is especially useful for studying human navigation behavior considering that the experimental findings did not seem very novel or interesting. The conclusions lists some simple A-B tests that *could* be done but it is unclear to me if these really benefit that strongly from using the proposed flow graph. I do not feel qualified to evaluate the non-ML merit of this work, but it begs the question if ICLR is the right venue.

[1*] Thrun, Sebastian. "Probabilistic robotics." Communications of the ACM 45.3 (2002): 52-57.

**Questions:**

Can you motivate why this model is a sufficiently useful representation of human navigation given the limitations I listed above?

---

### Note · Authors · 2024-11-25

**Comment:**

Thank you for your time and insightful feedback on our paper. We greatly appreciate the time it took to read and offer comments and references. We have decided the paper needs significant revisions to better emphasize its contributions and align with its goals. As such, we will be reorganizing the work and withdrawing it for now. We deeply value your input and hope to use it to make the paper stronger in future. Thank you again!

**Withdrawal Confirmation:**

I have read and agree with the venue's withdrawal policy on behalf of myself and my co-authors.